# Impact of Maternal Nutrition and Perinatal Factors on Breast Milk Composition after Premature Delivery

**DOI:** 10.3390/nu11020366

**Published:** 2019-02-10

**Authors:** Jean-Michel Hascoët, Martine Chauvin, Christine Pierret, Sébastien Skweres, Louis-Dominique Van Egroo, Carole Rougé, Patricia Franck

**Affiliations:** 1Department of Neonatology, Maternite Regionale, CHRU Nancy, 54035 Nancy, France; sebastienskweres@hotmail.fr; 2DevAH, Lorraine University, 54500 Vandoeuvre les Nancy, France; 3Dietetic and Nutrition Unit, CHRU Nancy, 54035 Nancy, France; m.chauvin@chru-nancy.fr (M.C.); c.pierret@chru-nancy.fr (C.P.); 4Saint Cloud Hospital, 92210 Paris, France; louisdo.vanegroo@club-internet.fr; 5Bledina Limonest, 69410 Champagne-au-Mont-d’Or, France; carole.rouge@danone.com; 6Biology Laboratory, CHRU Nancy, 54035 Nancy, France; p.franck@chru-nancy.fr

**Keywords:** maternal nutrition, breast milk, premature delivery, milk composition

## Abstract

(1) Background: Premature infants require mothers’ milk fortification to meet nutrition needs, but breast milk composition may be variable, leading to the risk of inadequate nutrition. We aimed at determining the factors influencing mothers’ milk macronutrients. (2) Methods: Milk samples were analyzed for the first five weeks after premature delivery by infrared spectroscopy. Mothers’ nutritional intake data were obtained during standardized interviews with dieticians, and then analyzed with reference software. (3) Results: The composition of 367 milk samples from 81 mothers was (median (range) g/100 mL): carbohydrates 6.8 (4.4–7.3), lipids 3.4 (1.3–6.4), proteins 1.3 (0.1–3.1). There was a relationship between milk composition and mothers’ carbohydrates intake only (*r* = 0.164; *p* < 0.01). Postnatal age was correlated with milk proteins (*r* = −0.505; *p* < 0.001) and carbohydrates (*r* = +0.202, *p* < 0.001). Multiple linear regression analyses showed (coefficient) a relationship between milk proteins *r* = 0.547 and postnatal age (−0.028), carbohydrate intake (+0.449), and the absence of maturation (−0.066); associations were also found among milk lipids *r* = 0.295, carbohydrate intake (+1.279), and smoking (−0.557). Finally, there was a relationship among the concentration of milk carbohydrates *r* = 0.266, postnatal age (+0.012), and smoking (−0.167). (4) Conclusions: The variability of mothers’ milk composition is differentially associated for each macronutrient with maternal carbohydrate intake, antenatal steroids, smoking, and postnatal age. Improvement in milk composition could be achieved by the modification of these related factors.

## 1. Introduction

Premature infants require mothers’ milk fortification to meet their nutrition needs [1]. This fortification is usually standardized using an assumed macronutrient milk composition [2]. However, studies have suggested that breast milk composition variability may be much wider than expected [3,4], leading to inadequate newborn nutrition. McLeod et al. performed a survey of protein and energy intake by milk analysis within the first 28 days of life in 63 infants born before 33 weeks’ gestation to assess their effect on growth [3]. Their results show that breast milk composition vary for all of the macronutrients with median protein concentrations of 16.6 g/L ranging from 13.4 g/L to 27.6 g/L, and median caloric intake of 73.3 Kcal/100 mL ranging from 63 Kcal/100 mL to 93 Kcal/100 mL. Of note, actual protein intake was correlated with infants’ growth. The authors concluded that preterm milk composition is very variable, and routine fortification using assumed averaged composition may result in inadequate nutrition with slower weight gain, as observed in their study [3].

Inadequate nutrition could indeed explain in part the postnatal growth restriction observed in many premature infants [5,6]. We aimed at determining factors, including mothers’ nutritional intake, which may be associated and explain breast milk macronutrient variability after premature delivery before 34 weeks’ gestational age. 

## 2. Materials and Methods

This is an observational study using milk bank data. In our level III Maternity Hospital, mothers’ own milk is pasteurized for premature infants. After pasteurization, we routinely analyze breast milk composition to verify the appropriateness of standardized fortification. For the purpose of the study, we collected data for all of the milk batches that each mother provided to the milk bank regardless of the time of the milk collection to evaluate the milk composition variation throughout the first five weeks of lactation. Each batch is a pool of one to three days, depending on the volume collected and frozen by each mother at home. Once a volume of at least 500 mL was collected, the mother would hand over the frozen batch to the milk bank for pasteurization. Hand-over occurred daily to twice a week.

Mothers who delivered a premature infant before 34 weeks’ gestation at our unit were enrolled within five days after delivery when they announced their choice for breastfeeding. Mothers’ dietary preference and nutritional intake data were obtained during personalized interviews with experienced dieticians. They used standardized questionnaires based on validated documents in the National French survey. The dieticians recalled dietary intake information from the two weeks prior to the interview. Perinatal data were prospectively collected at the time of the interview and from the mothers’ file. We analyzed maternal diet and macronutrient intake from the recall data averaged per day with appropriate software (Geni^®^ V7.0, Micro6, Villers les Nancy-F, France).

For milk analysis, we used the same protocol as described in the literature [7,8] and advised by the manufacturer. In short, mothers milk pools over one to three days were delivered frozen (−20 °C) to the milk bank, thawed, pasteurized, aliquoted in bottles of 50 mL, and stored at −80 °C. Two samples of one mL from two different bottles of each batch were analyzed after homogenization by ultrasound to compensate for milk thawing and verify the quality of the homogenization. We used infrared spectroscopy pre-calibrated against a chemical reference with analytical accuracy <0.1 g/100 mL (Miris AB^®^ V3, Uppsala, Sweden). Since it has been shown that mid-infrared analyzers may require calibration adjustment, we verified the calibration daily to ensure appropriate measures [9]. The samples were rejected if variability was over 10%; otherwise, the average of the two values was kept for further analysis.

We started our statistical analyses by power calculation relying upon McLeod [3]. We calculated that to demonstrate a significant relationship with a coefficient of at least *r* = 0.550, which is considered clinically significant, with an alpha risk of 0.016 (Bonferroni correction for the three nutrients) and a power of 0.90, 78 mothers with at least two samples would be needed (Power and Precision™ V4, Biostat Inc., Englewood, NJ, USA, 2001). Normally distributed data, assessed by a Shapiro–Wilk test of normality, are presented as mean values with standard deviation (SD), the median, and the interquartile range (IQR); non-normally distributed data are presented as medians with IQR only. To evaluate the differences between groups, we used the Student *t*-test for continuous variables and the Chi-squared test or Fisher exact test when appropriate for categorical variables. For continuous variables not normally distributed, we used the Mann–Whitney U test. To determine which variables were associated with milk composition, we performed a bivariate analysis, and then a stepwise multiple linear regression, including all of the variables associated with milk composition in bivariate analysis, with a tolerance set at 10^−5^ with the probability to remove the set at 0.15 and a confidence interval of 0.95. Observed differences were considered statistically significant if *p* < 0.05. Statistical analysis was performed with SYSTAT 12 software (2007, Systat Software Inc., San Jose, CA, USA).

The study has been approved by our Institutional Ethics and Review Board on 9 March 2013 (Number: MRU13-02).

## 3. Results

### 3.1. Description of the Studied Population

From August 2013 to January 2014, 367 milk samples were obtained from 81 mothers (Figure 1). Two to 10 batches were collected from the mothers (median (IQR): three (two to six)).

The mothers involved in the study were 29 years old (19–42) (median (range)). Their average height was 1.64 m (1.53–1.85) for a weight of 63 kg (42–110) and a body mass index (BMI) of 23.2 (16.4–43). Weight gain during pregnancy ranged from 0 to 30 kg (mean 10.2 kg). The age of delivery was 31 weeks’ gestational age (24–34). Twenty (25%) mothers smoked during pregnancy; 63 (78%) were single pregnancies, and 19 (23%) presented with toxemia. Prenatal maturation with corticosteroids was achieved in 37 (46%) mothers and partial maturation in 33 (41%). Vaginal delivery occurred in 42 mothers (52%) and cesarean section occurred in 39 (48%).

Forty-nine (60%) newborns were males; neonatal adaptation was good with an Apgar score above six at one and five minutes for all infants. Their mean birth weight was 1523 ± 512 g (median (range) = 1460 (600–2500) g).

### 3.2. Maternal Nutritional Intake and Milk Composition

Maternal nutritional intake was 2169 ± 562 Kcal/day (2146 (1197–3628)) with 88 ± 28 g/day (88 (40–213)) of fat intake, 86 ± 20 g/day (87 (40–160)) of protein intake, and 257 ± 81 g/day (247 (103–533)) of overall carbohydrate intake.

The global milk sample composition was (median [range]/100 mL): carbohydrates 6.8 g (4.4–7.3), lipids 3.4 g (1.3–6.4), proteins 1.3 g (0.1–3.1). The correlation between mothers’ food intake and milk composition is shown in Table 1.

### 3.3. Perinatal Factors’ Effect on Milk Composition

#### 3.3.1. Postnatal Age Effect on Milk Composition

Weekly mean protein content significantly decreased for the first four weeks post-delivery, and then remained stable at week five. We observed a comparable but inverse evolution for carbohydrates, and there was no significant difference over the first five weeks after delivery for the fat content of mothers’ milk (Table 2).

We observed a significant linear regression for carbohydrate and an inverse correlation for protein contents, as shown in Figure 2.

#### 3.3.2. Perinatal Factors’ Impact on Milk Composition in Bivariate Analysis

There was no relationship between the milk composition and the mothers’ age, weight, height, or BMI before pregnancy. There was no relationship either for the mode of delivery, multiple pregnancies, toxemia, or gestational age at delivery. We observed a weak association between weight gain during pregnancy and milk lipid content (*r* = 0.117, *p* = 0.026). Finally, we observed a significant correlation between milk composition and antenatal steroid maturation, smoking during pregnancy, and an inverse relationship with the infants’ birth weight. Detailed data are shown in Table 3.

#### 3.3.3. Stepwise Multivariate Regression Analysis of Factors Associated with Milk Composition in Bivariate Analysis

All of the factors showing a significant association with mothers’ milk content in bivariate analysis were included in the model presented in Table 4.

## 4. Discussion

Our study confirms that breast milk composition in macronutrients has a large variability, as suggested in previous studies [3,4]. Therefore, fortification of the assumed averaged macronutrient in milk leads to inadequate nutritional intake for most premature infants. Our data show that indeed, maternal nutrition may influence breast milk macronutrient composition in mothers who delivered prematurely. Our data also show that only overall carbohydrate is positively correlated with protein, fat, and caloric density. To our knowledge, this observation has not been yet reported. In their study comparing the breast milk nitrogen content of mothers from two different Chinese areas, Zhao et al. found no significant difference in 18 studied amino acids, despite significant lower protein intake in one of the studied areas [10]. This is consistent with our results. Likewise, in their study on maternal supplementation with omega-3 precursors, Mazurier et al. showed a qualitative alteration with increased alpha-linoleic acid content, but no significant modification of the overall lipid concentration [11]. This observation also recalls that a true difference in the secretory activity of the mammary gland is not merely a difference in concentration. In a recent survey on food and nutrient intake of women in France, Hebel et al. showed that few lactating mothers met the nutritional guidelines, and may therefore be at risk of food and nutrient inadequacies [12]. One could speculate that improving the maternal nutritional balance, especially an appropriate overall carbohydrate intake, might contribute to improved milk composition for preterm infant feeding.

Longitudinal analysis of milk composition showed a significant decrease in milk protein content with an inverse correlation for carbohydrate and a stable lipid concentration over the first four weeks after delivery. These data are consistent with the publication of Maly et al. [8], who found the same evolution for the three macronutrients and Mahakan et al. [4], who found a 50% decrease in protein concentrations with a 30% increase in carbohydrate concentrations over the first 28 days after delivery. From his review analysis on human milk in premature infants [13], which included the studies Charpak et al. [14] and Bauer et al. [15], Underwood evaluated the changes in milk composition over eight weeks from delivery, and found the same results as in our study over this longer assessment period. Yoneyama et al. suggested that there may be a negative correlation between protein content and milk volume and between early and mature milk [16]. This observation may explain why breastfeeding infants adjust their volume intake in relation with milk caloric density and protein content. However, when feeding premature neonates, the amount of milk is usually fixed to a target volume [17]. Therefore, protein content evolution is important to take into account when considering the fortification of human milk for premature infants.

Few data are available for the impact of perinatal factors on milk composition. In our study, we did not find any significant association among the mothers’ age, weight, height, or BMI before pregnancy. In their study on maternal nutrition and body composition during breast feeding, Bzikowska-Jura et al. [7] found a variance in milk fat content related to BMI. However, they studied the actual BMI at three time points, while we differentiated between BMI before pregnancy and weight gain during pregnancy, which was indeed associated with milk fat content. Thus, our results are consistent with published data, but suggest that it is not BMI per se, but rather weight gain during pregnancy that may be associated with milk composition. Also, there was no relationship for the mode of delivery, multiple pregnancies, or toxemia. As in our study, Maly et al. [8] showed no effect of the degree of prematurity at delivery on milk composition. However, we found a moderate effect of birth weight in bivariate analysis, but not after adjusting for other confounding factors in multivariate analysis (data not shown). Finally, we showed that breast milk protein concentrations were positively correlated with mothers’ overall carbohydrate intake, and negatively correlated with the duration of lactation from birth onwards, and the absence of steroid maturation; breast milk carbohydrate concentrations were positively correlated with the duration of lactation, and negatively correlated with smoking, as shown in the study of Bachour et al. [18], whereas breast milk lipid concentrations were positively correlated with mothers’ carbohydrate intake and negatively correlated with smoking.

Our study has strength and limitations. Our strength relies upon the number of samples, the blinded collection of the data, and the reproducibility of milk content measurements with a rejection when the control variability was above 0.10. Also, the dietary questionnaire over two weeks recall, which was given by experienced dieticians with an illustrated standardized catalog for the amount of food, allowed an appropriate evaluation of the averaged usual diet of the mothers. However, this questionnaire was given two weeks from the interview, and may not strictly reflect the diet of the lactating mother at the time of milk collection. Anyhow, one may speculate that the level of macronutrients intake would be only slightly modified from the routine diet of the patients. We did not find a significant difference from the results with sugar, fibers, or overall carbohydrate; therefore, we presented only the results with overall carbohydrate, but this would be confirmed by further targeting the study. Finally, because this is an observational study, we were not able to standardize the time that the mothers would hand over their milk collected at home to the milk bank, nor the number of collection days within each pool. However, this would only vary from one to three days, and we longitudinally evaluated the results over five weeks. Thus, even though this study does not really present an individual longitudinal analysis, the linear regressions that were observed for the overall population and the multivariate analysis, taking time as a confounding factor, are likely to allow a good reliability for our findings.

In conclusion, our study confirms that human milk macronutrients composition has a wide variability. This variability is differentially associated for each macronutrient, and associated with maternal carbohydrates intake, antenatal steroids, smoking, and the delay from delivery. To improve mothers’ milk composition to values closer to the one needed for achieving infants’ nutrition needs [19], one could aim for improving the prenatal nutritional balance in pregnant women, particularly for carbohydrate intake, supporting an appropriate weight gain during pregnancy, and advising stopping smoking. In the case of expected prematurity, steroid maturation is also needed to improve milk composition. Ideally, breast milk composition would be regularly measured for individualized fortification to achieve the appropriate growth of preterm infants. However, when it is only possible to apply standard fortification, at least human milk composition in relation to lactation duration from birth should be taken into account. Targeting investigation should be performed and confirm the observed data.

## Figures and Tables

**Figure 1 nutrients-11-00366-f001:**
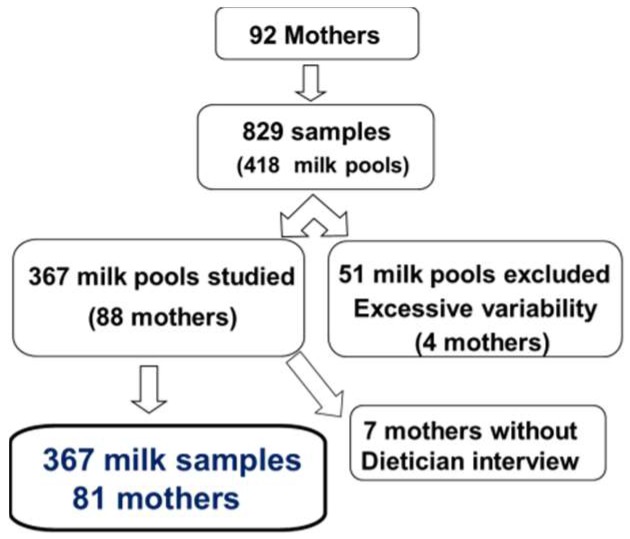
flow chart.

**Figure 2 nutrients-11-00366-f002:**
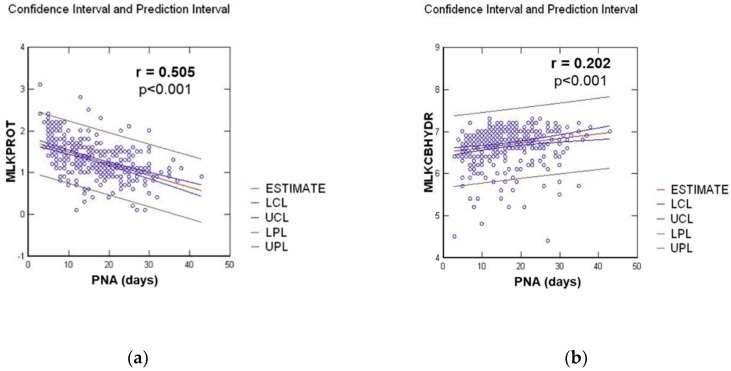
Linear regression between milk composition in g/100 mL and postnatal age (PNA) in days, for: (**a**) Protein milk content (MLKPROT); (**b**) Carbohydrate milk content (MLKCBHYD).

**Table 1 nutrients-11-00366-t001:** Linear regression between mothers’ food intake and milk composition (coefficient *r*).

Nutrients Intake	Milk Calories	Milk Proteins	Milk Lipids	Milk Carbohydrates
Log Energy	0.110 *	0.094	0.106 *	0.034
Protein	0.01	0.03	0.01	0.04
Fat	0.03	0.04	0.03	0.03
Carbohydrates	0.131 **	0.109 *	0.127 *	0.035

* *p* < 0.05; ** *p* < 0.01.

**Table 2 nutrients-11-00366-t002:** Average milk composition per postnatal week.

Week	1 (*n* = 52)	2 (*n* = 125)	3 (*n* = 91)	4 (*n* = 68)	5 (*n* = 31)
Protein	1.78 ± 0.39 *	1.40 ± 0.40 *	1.26 ± 0.34 *	1.08 ± 0.36 *	1.05 ± 0.40
Lipids	3.23 ± 0.80	3.58 ± 0.98	3.59 ± 0.97	3.41 ± 0.96	3.40 ± 1.06
CHO	6.50 ± 0.43 *	6.66 ± 0.38 *	6.70 ± 0.46 *	6.81 ± 0.44 *	6.75 ± 0.44

* *p* < 0.01.

**Table 3 nutrients-11-00366-t003:** Confounding perinatal factors for milk composition.

Milk Content (Mean, g/100 mL)	Smoking		Antenatal Steroids		Birth Weight (Linear Regression: r)
	Yes	No	Yes	No	
Lipids	3.10	3.59 *	3.6 *	3.1	0.082
Carbohydrates	6.57	6.71 *	6.7	6.6	−0.259 *
Protein	1.34	1.33	1.3	1.2	−0.318 *
Calories	62.8	67.8 *	67.6 *	61.9	−0.106

* *p* < 0.05.

**Table 4 nutrients-11-00366-t004:** Multivariate analysis of factors associated with mothers’ milk content.

Milk Content	Postnatal Age	Carbohydrate Intake	Smoking	No Steroids	Weight Gain
**Lipids** (*r*^2^ = 0.087)	NS	1.279 *	−0.557 *	NS	NS
**Carbohydrates** (*r*^2^ = 0.071)	0.012 *	NS	−0.167 *	NS	NS
**Protein** (*r*^2^ = 0.299)	−0.028 *	0.449 *	NS	−0.066 *	NS
**Calories** (*r*^2^ = 0.101)	NS	14.053 *	−5.901 *	NS	NS

* *p* < 0.05; NS: non-significant.

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
