# Peer review of "Impact of Maternal Nutrition and Perinatal Factors on Breast Milk Composition after Premature Delivery"

_nutrients, 2019, doi:10.3390/nu11020366_

Round 1

Reviewer 1 Report

There are  two points that are worth to be discussed:

A negative correlation between protein content and milk volumes has been reported.

A comparison of the relative proportions of the main ingredients of the fluid secreted by the mammary gland before and after parturition shows that a true difference in secretory activity is not merely a difference in concentration. Epitelial tight junctions have a role in this mechanism.

Continuous monitoring of macronutrient concentration of milk samples is mandatory to collect milk with a high protein content, especially suitable for preterm infants.

The debate between tailored and standard supplemmentation continues.

Author Response

There are two points that are worth to be discussed:

A negative correlation between protein content and milk volumes has been reported.

 We agree that a negative correlation between protein content and milk volume has been reported (e.g. Yoneyama et al.). However, when feeding premature neonates, the amount of milk is usually fixed to a target volume (e.g. about 155±5 ml/Kg/d [see Rochow et al.]). Therefore, protein concentration is important to take into account when considering fortification of human milk for premature infants.

This point has been added to the Discussion section of the manuscript.

A comparison of the relative proportions of the main ingredients of the fluid secreted by the mammary gland before and after parturition shows that a true difference in secretory activity is not merely a difference in concentration. Epitelial tight junctions have a role in this mechanism.

 We agree with this observation and already addressed this point in describing Mazurier et al study on maternal supplementation with omega-3 precursor. The aim of our study was to evaluate quantitative variations in macronutrient to emphasize the interest of monitoring macronutrient content. We did not study qualitative variations in secretory activity of the mammary gland. Anyhow, we added a comment on this point in the Discussion section of the manuscript.

Continuous monitoring of macronutrient concentration of milk samples is mandatory to collect milk with a high protein content, especially suitable for preterm infants.

 In our study, we focused on mothers’ own milk which is the recommendation for infants’ feeding. Thus, the monitoring of macronutrient is rather useful for supplementation to adjust composition to target values (Rochow et al) than select milk with more appropriate composition. However, in our study, we speculate that smoking cessation, maternal nutritional balance improvement and prenatal maturation could lead to improved composition. Targeting investigation should be performed and confirm the observed data.

The debate between tailored and standard supplementation continues.

 We agree that this is an ongoing debate and that is why we propose in our conclusion a balanced solution between ideal theoretical tailored supplementation, which is very difficult and often impossible to achieve in practice, and standardized fortification which leads to inappropriate nutrition in most of the cases.

We thank Reviewer 1 for her(his) valuable comments.

Reviewer 2 Report

Overall the study is solid but on page 6 line 206-208 you state "To improve mother's milk composition...." I am not sure I saw where you give a standard or "reference" value to which you compared it to (Perhaps I missed it).

Another issue is WHY did you pool the samples? In pooling did everybody give the same amount of pooled milk or for example did subject 1 give 200 ml while another gave 50 ml or whatever?

Author Response

Overall the study is solid but on page 6 line 206-208 you state "To improve mother's milk composition...." I am not sure I saw where you give a standard or "reference" value to which you compared it to (Perhaps I missed it).

 Your comment is totally appropriate: because mothers milk composition is so variable from mother to mother, we actually only have averaged values (see Gidrewicz et al). The point would be to improve milk composition to values closer to the one needed for achieving infants’ nutrition needs. We added a reference and modified the sentence for the sake of clarity.

Another issue is WHY did you pool the samples? In pooling did everybody give the same amount of pooled milk or for example did subject 1 give 200 ml while another gave 50 ml or whatever?

 Actually, we did not pool the samples given by the mothers but, as stated in the limitation section of the Discussion, we were not able to standardized the time the mothers would hand over their milk, collected at home, which were pooled from 1 to 3 days. Because this was an observational study, we could not modify the routine process and we agree that this is a limitation as we pointed out in our manuscript.

We thank Reviewer 2 for her(his) valuable comments.

Round 2

Reviewer 2 Report

Thank you for addressing issues I brought up.